# Stage-specific effects of Notch activation during skeletal myogenesis

**Pengpeng Bi[1†], Feng Yue[1†], Yusuke Sato[1], Sara Wirbisky[2], Weiyi Liu[1], Tizhong Shan[1], Yefei Wen[1], Daoguo Zhou[3], Jennifer Freeman[2,4], Shihuan Kuang[1,4]***

[1]Department of Animal Sciences, Purdue University, West Lafayette, United States; [2]School of Health Sciences, Purdue University, West Lafayette, United States; [3]Department of Biological Sciences, Purdue University, West Lafayette, United States; [4]Center for Cancer Research, Purdue University, West Lafayette, United States

*For correspondence: skuang@purdue.edu

[†]These authors contributed equally to this work

**Abstract** Skeletal myogenesis involves sequential activation, proliferation, self-renewal/differentiation and fusion of myogenic stem cells (satellite cells). Notch signaling is known to be essential for the maintenance of satellite cells, but its function in late-stage myogenesis, i.e. post-differentiation myocytes and post-fusion myotubes, is unknown. Using stage-specific Cre alleles, we uncovered distinct roles of Notch1 in mononucleated myocytes and multinucleated myotubes. Specifically, constitutive Notch1 activation dedifferentiates myocytes into Pax7 quiescent satellite cells, leading to severe defects in muscle growth and regeneration, and postnatal lethality. By contrast, myotube-specific Notch1 activation improves the regeneration and exercise performance of aged and dystrophic muscles. Mechanistically, Notch1 activation in myotubes upregulates the expression of Notch ligands, which modulate Notch signaling in the adjacent satellite cells to enhance their regenerative capacity. These results highlight context-dependent effects of Notch activation during myogenesis, and demonstrate that Notch1 activity improves myotube's function as a stem cell niche.

## Introduction

Skeletal muscle accounts for approximately 40% of adult human body weight (*Yin et al., 2013*). Besides the motor function that is essential for life, muscle is also a key metabolic organ, responsible for 80–90% of postprandial insulin-stimulated glucose uptake. In addition, muscle intimately interplays with many other organs, and regulates their homeostasis and metabolism (*Baskin et al., 2015*). Therefore, proper muscle function is important for human health and life quality.

In adult skeletal muscles, resident stem cells called satellite cells (SCs) are indispensable for postnatal muscle growth and regeneration (*Le Grand and Rudnicki, 2007*; *Lepper et al., 2011*; *Murphy et al., 2011*; *Pawlikowski et al., 2015*; *Relaix and Zammit, 2012*; *Sambasivan et al., 2011*). SCs are wedged between the plasma membrane of muscle fiber (i.e., myofiber) and the basal lamina that surrounds the myofiber. The juxtaposition of SCs immediately identifies myofiber as an important regulator of SCs. Consistent with this notion, transplantation of myofibers carrying SCs, but not SCs alone, elicited life-long amelioration of aging-related muscle atrophy (*Hall et al., 2010*). Besides myofibers, SCs also actively interact with interstitial fibroblasts, preadipocytes, endothelial cells and macrophages, which together with myofibers constitute the unique microenvironment, or niche that regulates the homeostasis of SCs (*Arnold et al., 2007*; *Aurora and Olson, 2014*; *Christov et al., 2007*; *Joe et al., 2010*; *Liu et al., 2012b*; *Murphy et al., 2011*; *Sonnet et al., 2006*).

**eLife digest** Muscles do much more than enable the body to move; they are also important organs involved in the metabolism. Conditions ranging from muscular dystrophy to insulin resistance result from problems that affect muscle tissue. Hence, understanding the signaling mechanisms that regulate how muscles develop and work will be critical to improving muscle-related health conditions.

To form and repair muscles, muscle progenitor cells develop (or differentiate) into new muscle cells, which then fuse to form muscle fibers. A signaling pathway involving a protein known as Notch regulates how cells communicate during development, and has been shown to play a key role in muscle progenitor cells. However, it was not known what role Notch signaling plays in the differentiated muscle cells.

Bi, Yue et al. have now studied genetically modified mice in which Notch signaling could be manipulated in certain types of cells. In mice with increased Notch signaling in both their newly differentiated muscle cells and muscle fibers, any unfused muscle cells were forced to return to an undifferentiated state, a process called dedifferentiation. This led to the muscles wasting away and resulted in the mice dying young. By contrast, in mice that only experienced activated Notch signaling in their muscle fibers, no dedifferentiation was seen. However, aged and dystrophic muscles in these mice regained the ability to contract and regenerate.

Bi, Yue et al. hope that these findings will transform into new strategies to activate or inactivate Notch signaling at different stages of muscle development or regeneration. This could help to repair muscles under various disease conditions.

Age-related muscle wasting (sarcopenia) and degenerative muscular dystrophy are linked to decline of SCs' abundance and functionality, as a result of changes in niche function and intrinsic properties of SCs (*Conboy et al., 2005*; *Sacco and Puri, 2015*). As such, FGF2, p38, STAT3, p16, canonical Wnt and Notch signaling, and epigenetic status are identified as aberrantly dysregulated factors in aged SCs that account for impaired self-renewal of SCs during aging (*Bernet et al., 2014*; *Brack et al., 2007*; *Chakkalakal et al., 2012*; *Cosgrove et al., 2014*; *Liu et al., 2013*; *Price et al., 2014*; *Sousa-Victor et al., 2014*; *Tierney et al., 2014*).

Stepwise progression of myogenesis is tightly orchestrated by several key transcriptional factors (*Bentzinger et al., 2012*). Among them, Pax7 (Paired box 7) is the faithful marker for SCs, and also indispensable for proper postnatal muscle regeneration by controlling expansion and differentiation of SCs (*Kuang et al., 2006*; *Seale et al., 2000*; *von Maltzahn et al., 2013*). Activation of myogenesis relies on the expression of several members of the basic helix-loop-helix domain-containing myogenic regulatory factors: Myf5, Myod1 (also known as MyoD), Myf6 and myogenin (Myog) (*Bentzinger et al., 2012*). When stimulated, SCs break quiescence and express MyoD, which promotes S-phase entry and facilitates SCs activation (*Halevy et al., 2004*; *Olguin and Olwin, 2004*; *Zammit et al., 2004*; *Zhang et al., 2010*). In addition, MyoD and Myf5 act as upstream regulators of Myog and Myf6, which are required for terminal differentiation of myocytes (*Bentzinger et al., 2012*). Notably, certain microRNAs, epigenetics, serum response factor (SRF) and members of myocyte enhancer factor 2 (MEF2) are also fundamentally important for proper muscle development (*Cheung et al., 2012*; *Haberland et al., 2009*; *Li et al., 2005*; *Liu et al., 2014*; *2012a*; *Shenoy and Blelloch, 2014*; *Williams et al., 2009*), which eventually leads to expression of muscle tissue-specific genes such as muscle creatine kinase (MCK), myosin light chain (MLC) and myosin heavy chain (MHC) members.

The opposite process of differentiation, or dedifferentiation is a fascinating phenomenon that bears great therapeutic potentials by generating stem cell sources for tissue repair. In zebrafish and amphibians, dedifferentiation is an integral part of tissue regeneration (*Jopling et al., 2010*; *Morrison et al., 2006*). Specifically, in the newt, fragmentation and dedifferentiation of myofibers generate a pool of proliferating mononucleated cells that give rise to the skeletal muscle of a new limb (*Sandoval-Guzman et al., 2014*). However, in mammals, there is no evidence of dedifferentiation that occurs as a natural part of tissue regeneration. The underlying mechanism that determines

such dramatic inter-species difference is largely unknown. Intriguingly, recent studies reported that dedifferentiation can be induced in muscle cells of several genetically engineered mice (*Kubin et al., 2011*; *Pajcini et al., 2010*). For instance, concomitant inactivation of two tumor-suppressors Arf and Rb in mononucleated mouse myocytes led to loss of differentiation properties, cell cycle reentry and generation of cells that were capable to redifferentiate into skeletal muscles (*Pajcini et al., 2010*). Undoubtedly, better understanding of mechanisms underlying muscle dedifferentiation will allow the invention of future medicine to treat geriatric muscle disease, where the SC number is limited.

Notch signaling is an evolutionarily conserved pathway that plays crucial functions in organ development, tissue homeostasis, stem cell fate choice, metabolism and cancers (*Andersson et al., 2011*; *Bi and Kuang, 2015*; *Koch et al., 2013*; *Ranganathan et al., 2011*). Notch signaling transduction is initiated upon binding of a Notch receptor (Notch1-4) with a ligand (Dll1, Dll4, Jag1, Jag2) located on a neighbor cell (*Andersson et al., 2011*). Subsequently, Notch receptors are cleaved by several enzymes including γ–secretase, which releases the Notch intracellular domain (NICD). NICD then translocates to the nucleus, where it binds with Rbpj and other cofactors to activate the transcription of canonical Notch targets, including Hes and Hey family genes (*Kopan, 2012*).

In postnatal myogenesis, Notch signaling is indispensable for maintaining quiescence and self-renewal of SCs (*Bjornson et al., 2012*; *Fukada et al., 2011*; *Mourikis et al., 2012b*; *Vasyutina et al., 2007*). In addition, during development SCs rely on Notch signaling to acquire 'satellite' position aside of myofiber by controlling the assembly of basal lamina (*Brohl et al., 2012*). Furthermore, Notch is a potent inhibitor of muscle differentiation by inhibiting expression of MyoD (*Delfini et al., 2000*). As such, deletion of *Rbpj* or *Dll1* led to premature differentiation and depletion of SCs, thus a loss of muscle growth and severe muscle hypotrophy (*Schuster-Gossler et al., 2007*; *Vasyutina et al., 2007*). Despite of the wealth of knowledge about Notch signaling in SCs, its function in late-stage myogenesis is unknown. Here, we report that activation of Notch signaling dedifferentiates myocytes into Pax7-expressing SCs, leading to defective myogenesis. By contrast, activation of Notch signaling in post-fusion myotubes/myofibers restored the functionality and regenerative capacity of aged and dystrophic muscles.

## Results

### Sequential activation of *Myl1*^Cre and MCK-Cre in post-differentiation stages of myogenesis

*Myl1*^Cre (MLC-Cre) (*Bothe et al., 2000*) and MCK-Cre (*Bruning et al., 1998*) mouse strains are widely used tools to study gene function in post-differentiation muscle lineages, by inducing recombination that causes either overexpression or knockout of candidate genes. However, the spatiotemporal activation pattern of MLC-Cre and MCK-Cre has not been characterized. It is conceivable of an earlier expression of the MLC-Cre than MCK-Cre, based on the developmental expression of Myl1 and MCK at day-9.5 and day-13.0 post conception, respectively (*Lyons et al., 1991*, *1990*; *Zammit et al., 2008*). In order to directly visualize and trace the activation of these Cre lines, we generated Td-tomato (red fluorescence protein) reporters for MLC-Cre and MCK-Cre, termed MLC-Td and MCK-Td mice, respectively. First, we cultured myoblasts from 1-month old MLC-Td and MCK-Td mice, and found that none of the freshly isolated SCs expressed RFP. Then we expanded the cells and induced myogenic differentiation. Under growth conditions, RFP expression was first detected in a few MLC-Td but not MCK-Td myocytes that were Pax7⁻, MyoD⁻, MyoG⁻, but MF20⁺ (*Figure 1A* and *Figure 1—figure supplement 1*). Later on, RFP was readily detectable in fusion-competent MLC-Td myocytes immediately after switching to the differentiation medium (*Figure 1B*). By contrast, RFP was only detected in post-fusion myotubes starting from day-3 after induction of myogenic differentiation in MCK-Td cells (*Figure 1B*).

Next, we examined the Cre activation pattern in skeletal muscles in vivo. To unmask potential mosaic Cre activation in multinucleated myofiber that can't be distinguished by the Td reporter, we utilized the nTnG reporter (*Prigge et al., 2013*), in which nucleus-localized GFP (nG) or Td-tomato (nT) is expressed in the presence or absence of Cre, respectively. Intriguingly, 96% and 55% myonuclei (within myofibers outlined by dystrophin) were nG⁺ in cross sections of tibialis anterior (TA) muscles from postnatal day-12 MLC-nTnG and MCK-nTnG mice, respectively (*Figure 1C and D*). By 1.5-month old, all myonuclei (100%) became nG⁺ in both MLC-nTnG and MCK-nTnG mice (data not

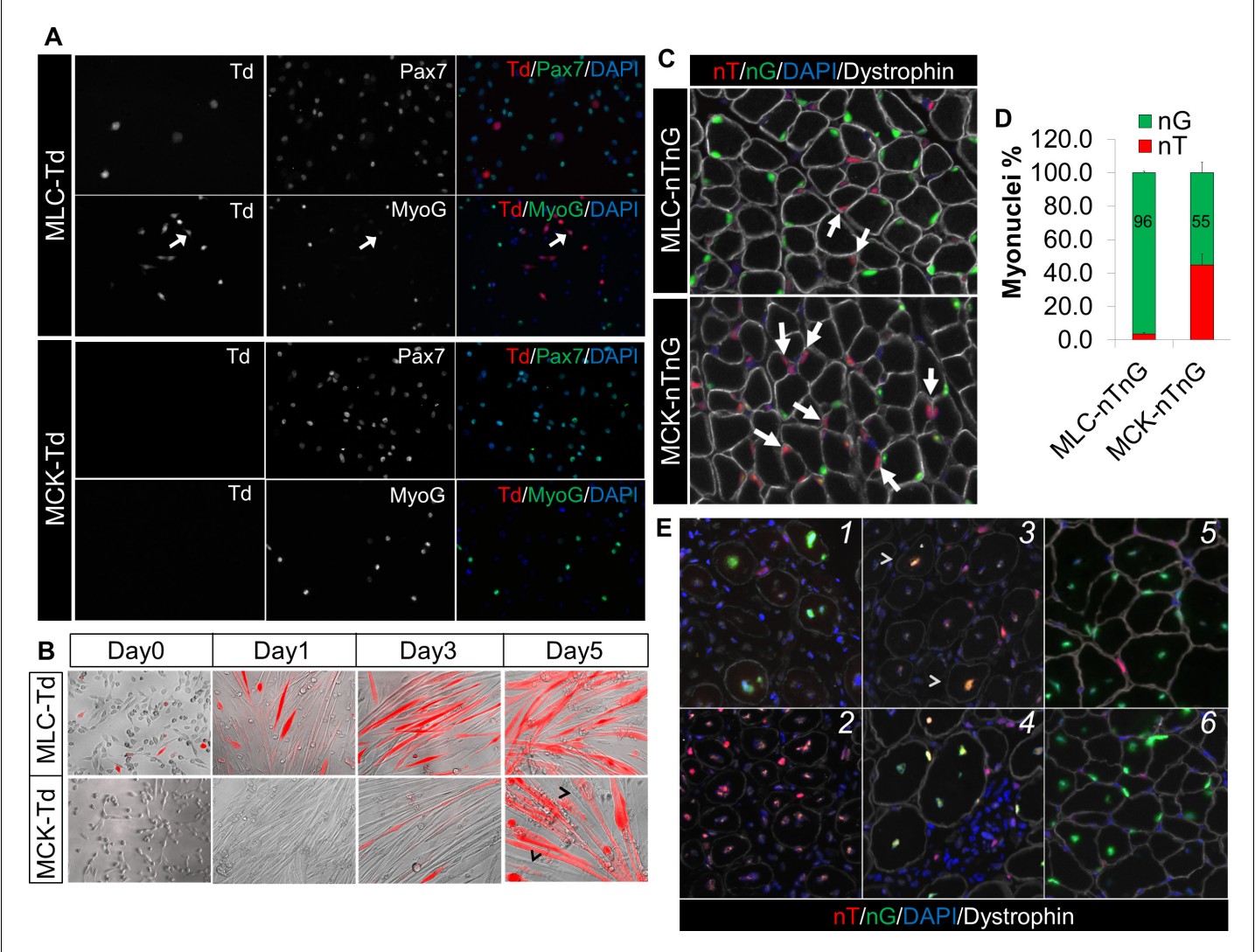

**Figure 1.** Sequential activation of *Myl1*[Cre] (MLC-Cre) and MCK-Cre in post-differentiation myocytes and post-fusion myotubes, respectively. (A,B) Immunofluorescence images of cultured myoblasts (A) and myocytes (B). Arrow in A points to a weak MyoG[+]/Td[+] cell. Arrowhead in B points to Td[−] myofibers. (C,D) Immunofluorescence images (C) of TA cross sections of postnatal day-12 mice, and myonuclei counting (D). Arrow points to nT myonucleus. (E) Immunofluorescence images of TA cross sections at different days post CTX injury (dpi): 1 (MLC-nTnG, 4 dpi), 2 (MCK-nTnG, 4 dpi), 3 (MCK-nTnG, 6 dpi), 4 (MCK-nTnG, 9 dpi), 5 (MLC-nTnG, 21 dpi), 6 (MCK-nTnG, 21 dpi). Arrowhead in panel 3 points to the nT/nG myofiber.

The following figure supplement is available for figure 1:

**Figure supplement 1.** Expression of myogenic markers in MLC-Td myoblasts.

shown), coinciding with the completion of the myonuclei addition in growing muscles (*White et al., 2010*).

We further examined the nT and nG expression at different stages of muscle regeneration after cardiotoxin (CTX) induced muscle injury. Consistently, we detected massive nG[+] myonuclei 4 days post injury (dpi) in MLC-nTnG mice (*Figure 1E1*). By contrast, MCK-nTnG myofibers didn't express nG at 4 dpi (*Figure 1E2*), but started to express nG at 6 dpi (*Figure 1E3*), and peaked at 9 dpi (*Figure 1E4*). Eventually, myonuclei in both MCK-nTnG and MLC-nTnG muscles were predominantly nG[+] at 21 dpi (*Figure 1E5 and E6*), which marks the completion of muscle regeneration. In summary, these in vitro and in vivo genetic mapping results reveal sequential activation of MLC-Cre and

MCK-Cre along myogenesis, with earliest MLC-Cre activation detected in mononuclear myocytes and MCK-Cre detected only in multinucleated myotubes (myofibers).

## Defective muscle growth and exercise performance in MLC-N1ICD mice

To investigate the role of Notch signaling in post-differentiation myocytes, we generated the *Myl1-Cre/Rosa26(Gt)Sor*[N1ICD] (abbreviated as MLC-N1ICD) mouse model, in which the expression of constitutively active Notch1-ICD (N1ICD) is driven by *Myl1*[Cre] (**Figure 2A**). The MLC-N1ICD mice were born at an expected Mendelian ratio with normal body weight (data not shown). Adult MLC-N1ICD mice expressed higher levels of *Notch1*[ICD] and its target genes, including *Hes1*, *Hey1* and *Heyl*, in both fast (gastroceminus, **Figure 2B**) and slow (soleus, **Figure 2C**) muscles, confirming activation of Notch signaling. However, MLC-N1ICD mice showed less body weights at weaning (postnatal day-21), and less body weight gains and muscle mass in adulthood, compared with wildtype (WT) mice (**Figure 2D and E**). By 4-month of age, all MLC-N1ICD mice displayed prominent kyphosis (**Figure 2F**), which is a sign of severe muscle weakness and a hallmark of aging. As early phase of postnatal muscle growth is normally mediated by myonuclei addition, we quantified the myonucleus numbers of EDL myofibers, and found that MLC-N1ICD had only around half the myonuclei numbers of WT myofibers (**Figure 2G and H**). Consequently, MLC-N1ICD mice were subjected to premature

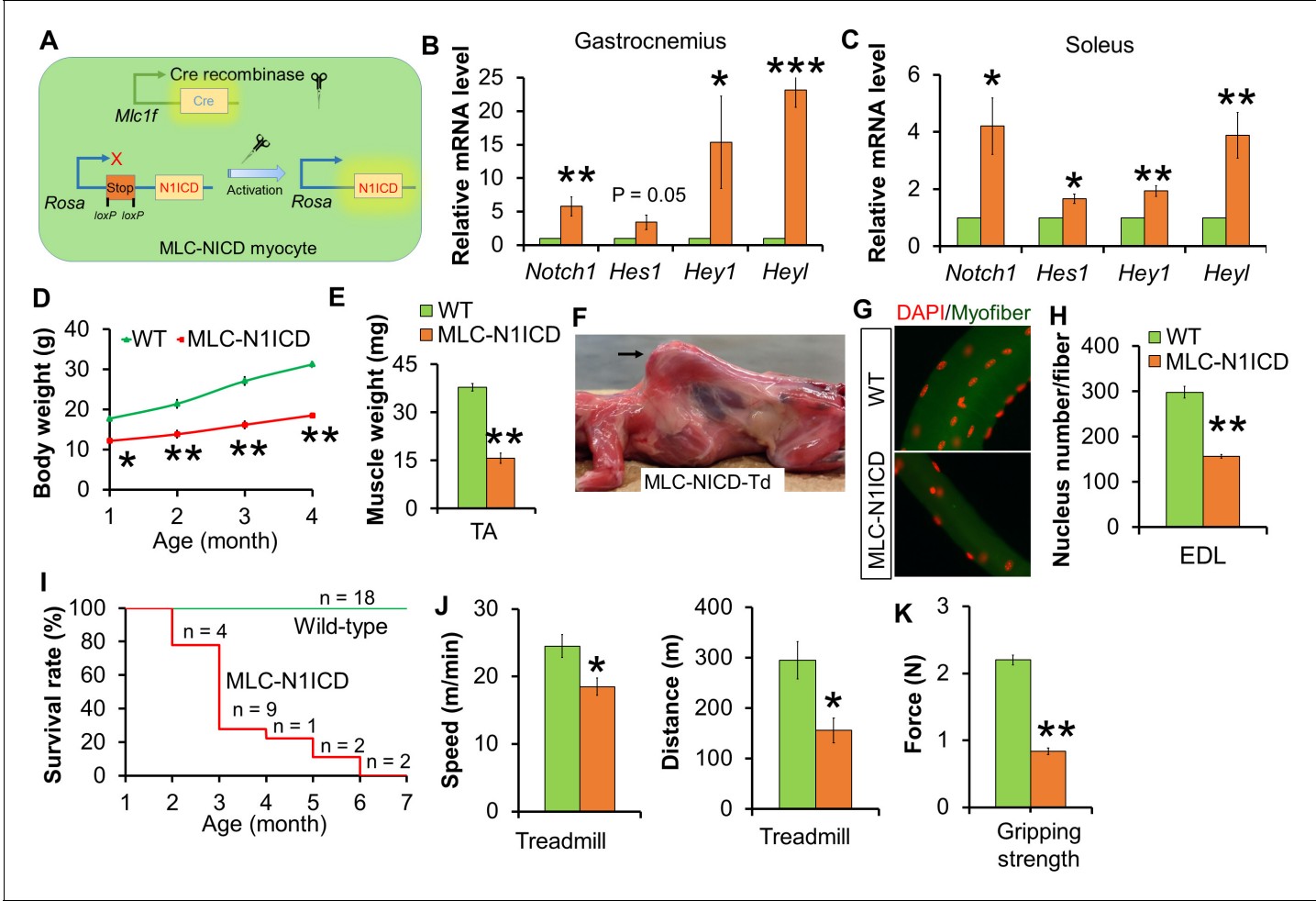

**Figure 2.** Muscle growth and motor-function defects of MLC-N1ICD mice. (A) Cartoon illustration of Notch activation by Cre in MLC-N1ICD mice. (B,C) Relative gene expression levels in fast (B, gastrocnemius, n = 3) and slow muscles (C, soleus, n = 4). (D) Growth curve of MLC-N1ICD mice (n = 3). (E) TA muscle weight (n = 4). (F) Image of MLC-N1ICD-Td mouse with kyphosis (arrow), note muscles are in red color as labeled by RFP (Td). (G,H) Immunofluorescence images of EDL fiber (G) and quantification of myonucleus numbers (H, n = 3). (I) Survive curve of MLC-N1ICD mice. (J) Exhaustive treadmill exercise test results (n = 4). (K) Gripping strength measurement result of limbs (n = 3). *p<0.05, **p<0.01. Bar graphs indicate mean SEM.

death starting from 2-month of age, and no MLC-N1ICD mice survived longer than 6 months (*Figure 2I*).

Although 1.5-month old MLC-N1ICD mice had smaller muscles, they moved normally in the cage. We utilized treadmill to assess if their exercise performance is impaired. Notably, MLC-N1ICD mice ran significantly more slowly and shorter distance compared with WT mice (*Figure 2J*). In addition, MLC-N1ICD mice showed lower gripping strength (*Figure 2K*). Together, these results reveal normal embryonic muscle development but dramatic defects in postnatal muscle growth and function in MLC-N1ICD mice.

## Muscle regeneration defect, and myocyte dedifferentiation in MLC-N1ICD mice

To evaluate the muscle regeneration potential of MLC-N1ICD mice, we injured TA muscles with CTX and evaluated the regeneration at 7 dpi. In non-CTX injected leg, the MLC-N1ICD myofibers are smaller than those of WT mice (*Figure 3A*, left; *Figure 2G*); with CTX injection, MLC-N1ICD showed drastically compromised muscle regeneration, evidenced by massive infiltration of non-muscle cells into TA (*Figure 3A*; right). We also performed Pax7 immunostaining to detect satellite cells. Unexpectedly, despite of severe regenerative defects, MLC-N1ICD muscles showed greatly increased number of Pax7$^+$ satellite cells, in both control and injured TA muscles (*Figure 3B*). Similarly, MLC-N1ICD EDL myofibers showed 10-times more Pax7$^+$ satellite cells than WT myofibers (*Figure 3C*). Consistently, Pax7 expression was significantly higher at both mRNA and protein levels in MLC-N1ICD muscles (*Figure 3D and E*). To determine whether these changes are caused by SC proliferation, we co-labeled MLC-N1ICD SCs with Ki67. However, none of the Pax7$^+$ cells expressed Ki67 (*Figure 3F*). Similarly, we didn't detect MyoD and MyoG expression in these cells (*Figure 3—figure supplement 1A*). As a positive control of staining, the cultured WT myoblasts were Ki67$^+$ and MyoD$^+$ (*Figure 3—figure supplement 2A,B*). These results indicate that these Pax7$^+$ cells are in a quiescent state. Indeed, we detected significantly higher expression levels of marker genes that are abundantly (*Cdh15*, encodes M-cadherin) or exclusively (calcitonin receptor, *Calcr*; teneurin transmembrane protein 4, *Tenm4*) expressed by quiescent SCs (*Figure 3G*) (*Fukada et al., 2007*). Similar changes were also observed in soleus muscles of MLC-N1ICD mice (*Figure 3—figure supplement 1B and C*).

As Notch signaling is required for maintaining the quiescence in SCs (*Bjornson et al., 2012*; *Mourikis et al., 2012b*), we hypothesized that these extra Pax7$^+$ SCs are dedifferentiated from MLC-Cre myocytes, as a consequence of cell-autonomous Notch activation. To test this hypothesis, we generated the Cre-inducible RFP (Td-tomato) reporter mice in combination with Notch activation, namely MLC-N1ICD-Td mice. Strikingly, 73% of the Pax7$^+$ SCs in MLC-N1ICD mice were labeled as Td$^+$ (*Figure 3H*; second row), demonstrating that they were previously differentiated (Myl1$^+$). As a control, no Pax7$^+$/Td$^+$ SC was detected in resting and CTX-injected muscles of adult MLC-Td mice (*Figure 3 H*; first row, *Figure 3—figure supplement 1D*), which is consistent with the earlier tracing result of cultured MLC-Td SCs (*Figure 1A*). In addition, in the CTX damaged muscles, we detected reduced numbers of Pax7$^+$/Ki67$^+$ cells in MLC-N1ICD versus WT mice (*Figure 3—figure supplement 2C,D*). In summary, Notch1 activation dedifferentiated MLC-Cre$^+$ myocytes into quiescent SCs.

## Improved muscle regeneration and exercise performance of aged MCK-N1ICD mice and dystrophic MCK-N1ICD-*mdx* mice

In parallel, to investigate the role of Notch signaling in post-fusion myofibers, we generated the MCK-Cre/*Rosa26(Gt)Sor*$^{N1ICD}$ (abbreviated as MCK-N1ICD) mouse model. MCK-N1ICD mice were born at normal Mendelian ratio and expressed higher levels of *Notch1* and Notch target genes, including *Hes1*, *Hes5*, *Hey1* and *Heyl* (*Figure 4—figure supplement 1A*). Compared to WT littermates, adult MCK-N1ICD mice didn't show any significant differences in body weight, myosin expression, neuromuscular junction morphology, denervation responses, exercise performance and gripping strength (*Figure 4—figure supplement 1B–H*).

In addition, adult MCK-N1ICD mice displayed normal muscle regeneration after a single episode of CTX injury (*Figure 4—figure supplement 2A*; first row). In response to multiple rounds of injuries induced by repetitive CTX injections, however, the MCK-N1ICD muscles regenerated much better

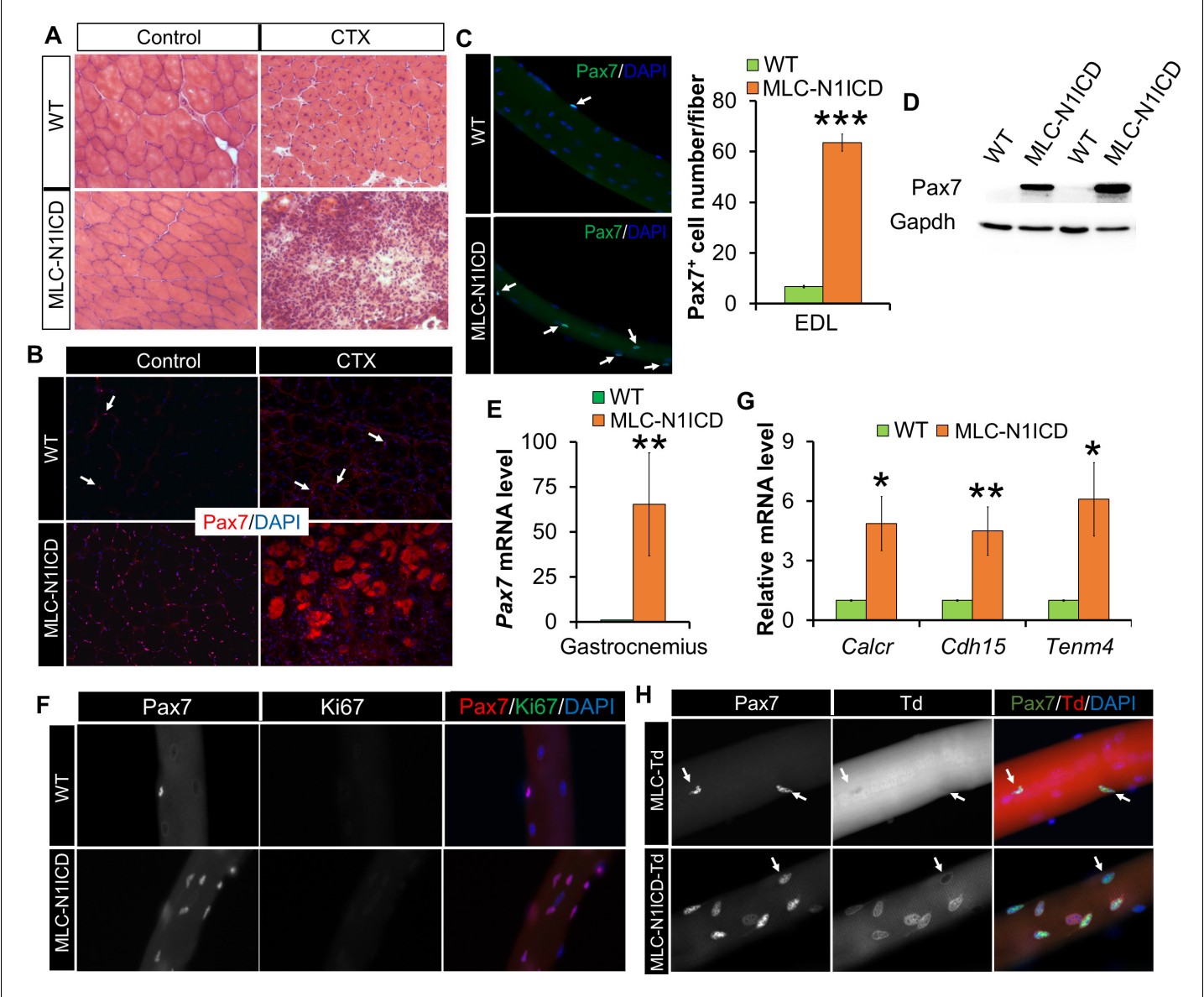

**Figure 3.** Muscle regeneration defect, and myocyte dedifferentiation in MLC-N1ICD mice. (**A**) H&E staining results of TA muscle cross sections. Right panels, 7 dpi. (**B**) Immunofluorescence images of TA muscle cross sections. Arrow points to Pax7+ cell in WT mice. (**C**) Immunofluorescence images (left) and quantification (right, n = 3) of Pax7+ cells (arrow) on EDL fibers. (**D**) Western blot results of protein extracts from non-injured muscle. (**E,G**) Relative expression of Pax7 (**E**, n = 3) and quiescent SC marker genes (**G**, n = 5) in gastrocnemius muscle. (**F,H**) Immunofluorescence images of EDL fibers. Note all Pax7+ cells are Ki67− in **F**; Note Pax7+/Td+ cells only appeared on MLC-N1ICD-Td but not on MLC-Td fibers in **H**, arrow points to Pax7+/Td− cells in H. *p<0.05, **p<0.01. Bar graphs indicate mean SEM.

The following figure supplements are available for figure 3:

**Figure supplement 1.** Myocytes dedifferentiation in MLC-N1ICD mice.

**Figure supplement 2.** Immunostaining results of myofibers, myoblasts and sections of CTX damaged TA muscles.

than the WT muscles, manifested by overall larger muscle volume (*Figure 4—figure supplement 2B*), appearance of larger regenerating myofibers and homogeneous regenerated area throughout the muscle (*Figure 4—figure supplement 2A*; second row). As aged muscles (>1-year old) expressed reduced levels of Notch receptors and Notch targets than young muscles (around 1

month old) (*Figure 4—figure supplement 2C and D*), we investigated if MCK-N1ICD improves muscle regeneration in aged mice. At 15-month old, MCK-N1ICD muscles regenerated more efficiently than those of WT littermates, evidenced by larger and more regenerating myofibers, reduced adipocyte infiltration (*Figure 4—figure supplement 2A*; third row), hallmarks of human sarcopenia (*Taaffe et al., 2009*). Moreover, the aged MCK-N1ICD mice achieved a higher maximum speed and longer running distance in the treadmill test (*Figure 4—figure supplement 2E*). Together, Notch1 activation driven by MCK-Cre improves muscle function and regeneration in aged mice.

We next asked if myofiber-specific activation of Notch1 improves muscle pathology in *mdx* mice, a widely used model for Duchenne Muscular Dystrophy (DMD) in humans. To achieve this goal, we generated MCK-N1ICD-*mdx* mice (short as N1ICD-*mdx*) and detected upregulation of Hes1 expression in the muscles of the N1ICD-*mdx* mice (*Figure 4A*), indicating Notch activation. A prominent feature of *mdx* mice is the continuous cycles of muscle regeneration and degeneration that lead to muscle pseudo-hypertrophy: larger but weaker muscles (*Chamberlain et al., 2007*). Interestingly, compared with littermate *mdx* mice, adult N1ICD-*mdx* mice showed 11% less body weight and 27% less muscle mass (*Figure 4B and C*). Such changes were not observed in 4-week old N1ICD-*mdx* mice (before pseudo-hypertrophy) and adult MCK-N1ICD mice (*Figure 4—figure supplement 1B*). Therefore, the body weight reduction phenotype of adult N1ICD-*mdx* mice is specific to the *mdx*, and coincides with the peak of muscular dystrophy in adult *mdx* mice (*Faber et al., 2014*). Consistently, H&E and immunostaining revealed the relatively smaller fiber size, but fewer centronuclear and necrotic IgG$^+$ myofibers in N1ICD-*mdx* mice, compared with *mdx* mice (*Figure 4D*; first row, E and F). Given this, we interpreted the reductions of muscle mass as a sign of less compensatory pseudo-hypertrophy and improved muscle function.

In addition, we performed CTX-induced muscle damage, followed by Evans blue dye (EBD) injection to label myofibers with dysfunctional and permeable membranes. Notably, the muscles of N1ICD-*mdx* mice showed much less EBD uptake than *mdx* muscles (*Figure 4G*). Consistently, there were fewer EBD$^+$myofibers in N1ICD-*mdx* than *mdx* muscles (*Figure 4H*). H&E staining revealed better muscle regeneration, characterized by more and larger myofibers, and less inflammatory infiltration in the CTX-injected TA muscles of N1ICD-*mdx* mice (*Figure 4D*; second row). Furthermore, adult N1ICD-*mdx* mice also exhibited better muscle exercise performance and stronger gripping strength than their littermate *mdx* mice (*Figure 4I–K*).

## N1ICD-*mdx* muscles showed gene expression signatures of healthy human muscles

To systematically evaluate the impact of myofiber-specific Notch1 activation on *mdx* muscles, we used microarray to survey the transcriptomes of N1ICD-*mdx* and *mdx* muscles. Among the total 882 differentially expressed genes (by ≥1.5–fold), 505 genes were downregulated and 377 genes were upregulated in N1ICD-*mdx*, compared with *mdx* muscles (*Figure 5—source data 1*). From the microarray, we identified *Gdf11* (Growth Differentiation Factor 11) as the highest upregulated gene in MCK-N1ICD muscles (*Figure 5—figure supplement 1A*). Consistently, we validated upregulation of *Gdf11* expression in all muscles examined in both MCK-N1ICD-*mdx* and MLC-N1ICD mice, compared with their littermate controls (*Figure 5—figure supplement 1B and C*). Considering the contrasting phenotypes of MLC-N1ICD and MCK-N1ICD mice, however, it is unlikely that Gdf11 directly mediates the effects of N1ICD overexpression in skeletal muscle function.

We performed Gene Set Enrichment Analysis (GSEA) (*Subramanian et al., 2005*) to interrogate the down- and up-regulated gene sets with a reference human DMD gene expression dataset from NCBI (GDS3027), which includes 14 healthy human muscle samples and 23 DMD human muscle samples (*Figure 5A*) (*Pescatori et al., 2007*). Commonly regulated genes in these murine and human datasets were assigned with enrichment scores that correlate to the statistical significance and fold changes of the genes' appearance in the human DMD expression dataset.

Strikingly, GSEA predicted significant overall enrichment (false discovery rate q-value = 0.02; familywise-error rate p-value = 0.009) for the downregulated N1ICD-*mdx* gene-set with human genes that were upregulated in DMD versus healthy human subjects (score, –1.40) (*Figure 5B*). In other words, the downregulated genes in N1ICD-*mdx* (versus *mdx*) muscles were expressed at higher levels in DMD (versus healthy) human muscles. Similarly, there is a strong trend that N1ICD-*mdx* upregulated genes were also positively correlated to genes that were expressed at higher levels in healthy human muscles (score = 1.08, q = 0.26, p = 0.13) (*Figure 5C*).

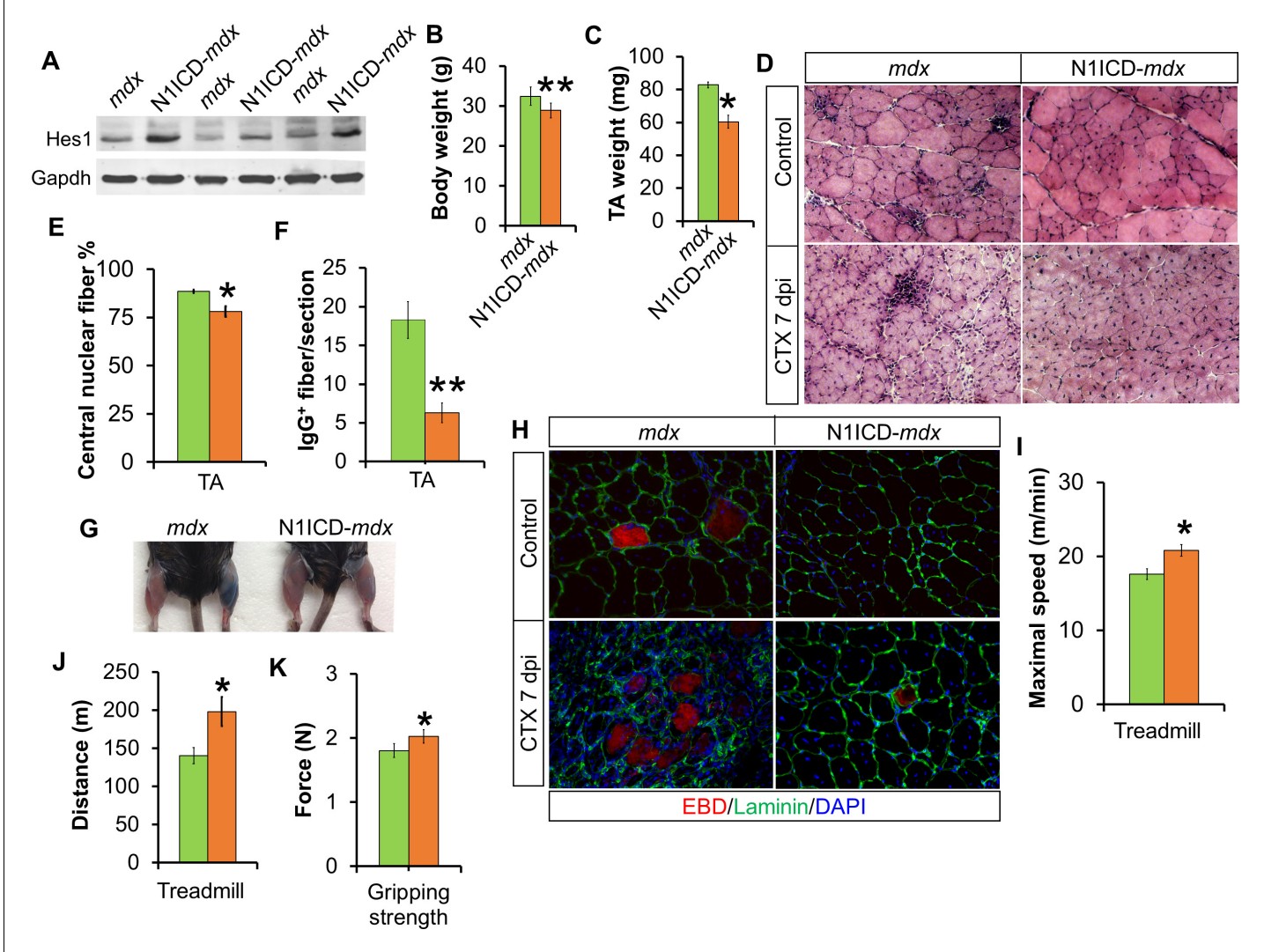

**Figure 4.** Improved muscle morphology, regeneration and exercise performance of adult MCK-N1ICD-*mdx* (short as N1ICD-*mdx*) mice. (**A**) Western blot results of Notch target gene Hes1 in muscle protein extracts. (**B,C**) Mice body weights (**B**, n = 4) and TA muscle weights (**C**, n = 3), to show less muscle pseudo-hypertrophy of N1ICD-*mdx*, versus *mdx* mice. (**D**) H&E staining results of TA muscle sections. (**E,F**) Quantification of central nuclei fiber ratio (**E**, n = 3) and IgG+ fiber numbers (**F**, n = 7) of non-CTX injected *mdx* and N1ICD-*mdx* mice. (**G**) Results of Evans blue dye (EBD) uptake by control (left leg) and 7 dpi CTX-injured muscles (right leg). (**H**) Immunofluorescence staining results of TA muscle cross sections. (**I,J**) Exhaustive treadmill exercise test results (n = 5). (**K**) Gripping strength measurement of limbs of adult mice (n = 15). *p<0.05, **p<0.01. Bar graphs indicate mean SEM.

The following figure supplements are available for figure 4:

**Figure supplement 1.** Normal muscle development, function and denervation response of MCK-N1ICD mice.

**Figure supplement 2.** Improved muscle regeneration and function of aged MCK-N1ICD mice.

---

Based on the GSEA result, we generated heatmaps for expression of the top-10 commonly down- and up-regulated genes in the murine and human arrays (*Figure 5D*). Consistent with their reduced degeneration, N1ICD-*mdx* muscles expressed higher levels of mature muscle specific *Myh1* (MHC-2x), and lower levels of regenerating muscle specific *Myh8* (*Figure 5D*). Strikingly, expression of *Spp1*, the biomarker of human DMD (*Pegoraro et al., 2011*), is also significantly lower in N1ICD-*mdx* muscles, than in *mdx* muscles (*Figure 5D*). Interestingly, the cell cycle inhibitor p21 (encoded by gene *Cdkn1a*), a critical modulator of senescence, and the muscle hypertrophy gene *Junb*

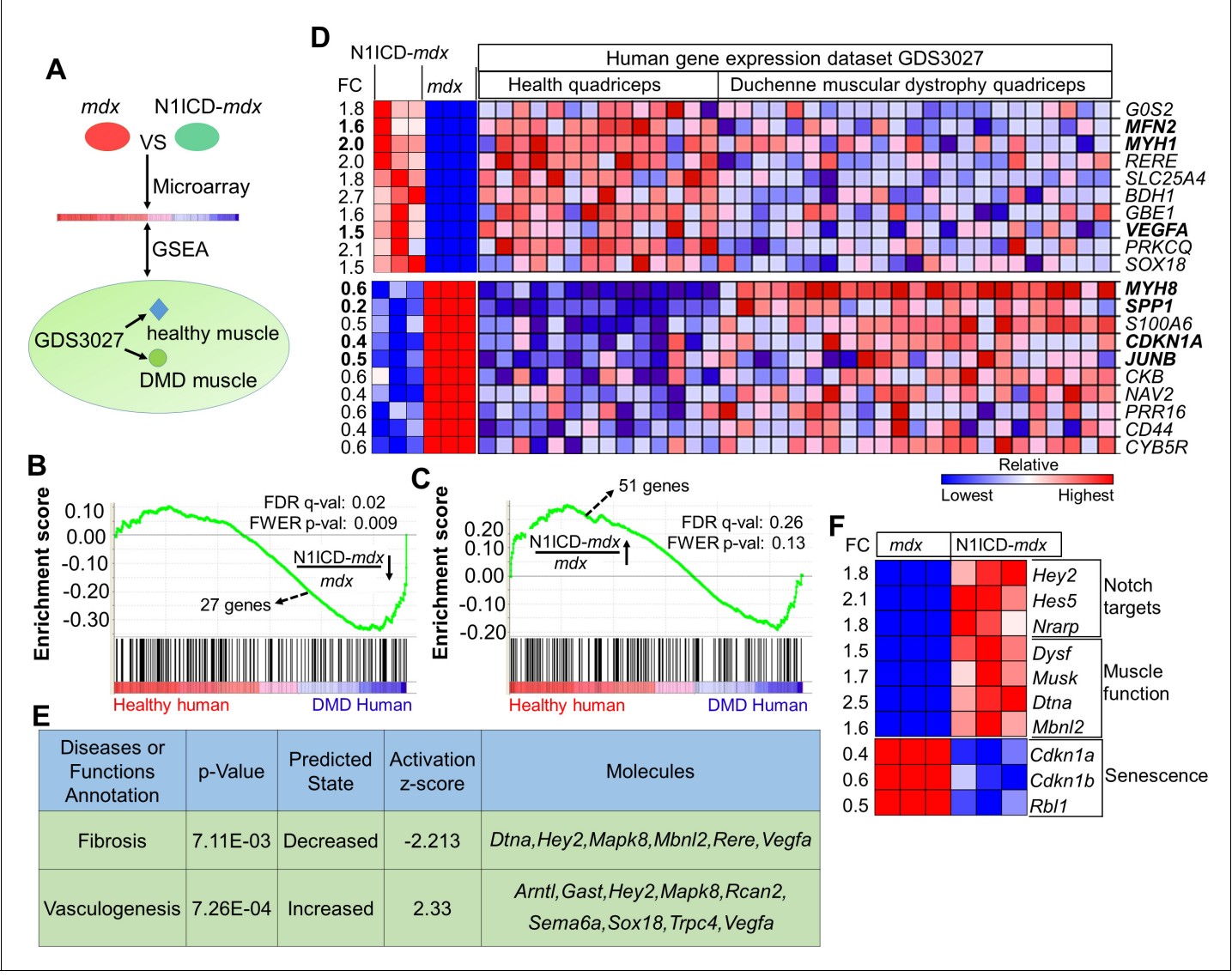

**Figure 5.** N1ICD-*mdx* muscle transcriptomes gained gene signatures enriched in healthy versus DMD human muscles. (**A**) Cartoon illustration of experiment design for (**B–D**). (**B,C**) Gene set enrichment plots from GSEA analysis of the normal human and DMD muscle gene expression database (GDS3027), interrogated with the down-regulated (**B**) and up-regulated (**C**) gene sets in N1ICD-*mdx* versus mdx muscles. FC, fold change. (**D**) Heatmap results of top 10 commonly up-regulated and down-regulated gene expression in N1ICD-*mdx* muscles and normal human muscles (GDS3027). (**E**) Ingenuity analysis of 51 genes that are commonly up-regulated in N1ICD-mdx muscles and normal human muscles (GDS3027). (**F**) Heatmap results of gene expression in the indicated pathways. FC, fold change in N1ICD-mdx relative to mdx muscles.

The following source data and figure supplement are available for figure 5:

**Source data 1.** Agilent microarray results showing genes up-regulated and down-regulated in N1ICD-*mdx* versus *mdx* muscles.
**Figure supplement 1.** Activation of Notch1 signaling upregulates Gdf11 expression in muscles.

(*Raffaello et al., 2010*) were significantly downregulated in N1ICD-*mdx* muscles and normal human muscles, compared with corresponding dystrophic muscles (*Figure 5D*). Both N1ICD-*mdx* and healthy human muscle showed upregulation of *Mfn2* (Mitofusin 2) (*Figure 5D*), which is involved in mitochondrial fusion, and associated with better muscle glucose metabolism and amelioration of muscle atrophy (*Chen et al., 2010*; *Sebastian et al., 2012*). In addition, upregulated in N1ICD-*mdx* samples was *Vegfa* (*Figure 5D*), which stimulates angiogenesis and improves vascular function.

Consistently, Ingenuity Pathway Analysis (IPA) of commonly up- and down-regulated genes predicted significantly elevated 'vasculogenesis', and decreased 'fibrosis' in N1ICD-*mdx* muscles (*Figure 5E*).

We also examined genes that were uniquely changed in our murine microarray. Notch target genes *Hey2*, *Hes5* and *Nrarp* were upregulated in N1ICD-*mdx* muscles (*Figure 5F*), confirming Notch activation. Interestingly, dysferlin (*dysf*) and α-dystrobrevin (*Dtna*), two components of dystrophin-glycoprotein complex essential for sarcolemma repair and neuromuscular junction maturation (*Bansal et al., 2003*; *Grady et al., 1999*), were significantly elevated in N1ICD-*mdx* muscles (*Figure 5F*). *Musk* (Muscle-specific kinase receptor) that regulates the development of the neuromuscular junction (*Perez-Garcia and Burden, 2012*) was also upregulated in N1ICD-*mdx* muscles (*Figure 5F*). Furthermore, a panel of stem cell senescence modulators, p21 (*Cdkn1a*), p27 (*Cdkn1b*) and *Rbl1* (*Cheung and Rando, 2013*; *Narita et al., 2003*; *Sousa-Victor et al., 2014*) were all significantly downregulated in N1ICD-*mdx* muscles (*Figure 5F*). In summary, these transcriptome analysis results support the phenotypic assays demonstrating the myofiber-specific Notch1 activation improves muscle function in dystrophic skeletal muscles.

## Notch1 activation in myofiber upregulates the expression of Dll/Jag ligands that modulates Notch signaling in adjacent satellite cells

Sarcopenia and muscle dystrophy are linked to decline of SC abundance and functionality, which is associated with repression of Notch signaling in SCs (*Conboy et al., 2003*; *Jiang et al., 2014*). Consistent with improved muscle regeneration, CTX-damaged 17-month old MCK-N1ICD muscles, and adult N1ICD-*mdx* muscles showed more Pax7$^+$ SCs (*Figure 6A–C*), and elevated expression of Pax7 than did *mdx* muscles (*Figure 6D*). To exclude the possibility that dedifferentiation of MCK-Cre$^+$ myocytes contribute to the increases of SCs, we again performed lineage tracing using MCK-Cre in combination with N1ICD and Td reporter mice. Staining of the EDL myofiber with Pax7 antibody didn't detect any Pax7$^+$/Td$^+$ cells (*Figure 6E*). Therefore, unlike the MLC-N1ICD mice, the increase of SCs number in MCK-N1ICD mice is not a result of myocyte dedifferentiation.

Considering the juxtaposed position of SCs on their host myofibers, Notch ligands on myofibers are presumably essential triggers of SC-intrinsic Notch signaling that is necessary for the quiescence and maintenance of SCs. Genetic mapping studies showed that the higher Jag1 expression is associated with better muscle phenotypes of dystrophic mice and dogs (*Vieira et al., 2015*). Notably, Jag1 overexpression rescued the muscular dystrophy phenotype of dystrophin-deficient zebrafish (*Vieira et al., 2015*). In skeletal muscle of N1ICD-*mdx* mice, we detected elevated expression of two Notch ligands, *Jag2* and *Dll4*, compared with *mdx* muscles (*Figure 6F*). Consistently, immunostaining of EDL myofibers revealed stronger Dll4 immunofluorescence in MCK-N1ICD than in WT myofibers (*Figure 6G and H*). Similar gene expression changes were detected in myofibers of MCK-N1ICD mice (*Figure 6—figure supplement 1A*).

To directly evaluate whether the increased ligand expression in myofibers subsequently activates Notch signaling in SCs, we generated MCK-N1ICD-CpGFP mice by breeding MCK-N1ICD with Notch reporter mice (CpGFP), in which expression of GFP was controlled by a promoter cassette containing the Rbpj-responsive element (*Jiang et al., 2014*; *Mizutani et al., 2007*). Immunostaining of EDL myofibers with Pax7 and GFP antibodies revealed that 41% SCs were GFP$^+$/Pax7$^+$ in MCK-N1ICD-CpGFP mice, compared with 26% GFP$^+$/Pax7$^+$ SCs in control CpGFP mice (*Figure 6I and J*). As a positive control, we observed stronger GFP fluorescence in myofibers of MCK-N1ICD-CpGFP mice, compared with those of CpGFP mice (*Figure 6I and K*), confirming Notch activation by N1ICD OE. As a negative control, no GFP$^+$ SC was observed on WT myofibers (*Figure 6I*; first row). Therefore, we hypothesize that MCK-N1ICD upregulates Dll4/Jag2 in myofibers, which in turn activates Notch signaling in juxtaposed SCs to regulate their self-renewal in response to aging and muscular dystrophy.

To address whether Notch1 signaling components are expressed in mature myofibers under normal physiological conditions, we examined EDL myofibers free of SCs and interstitial cells after trypsin digestion and extensive PBS washing (*Figure 6—figure supplement 2A*). We confirmed the complete removal of SCs as evidenced by the absence of Pax7$^+$ cells and Pax7 expression in the trypsin-stripped EDL myofibers (*Figure 6—figure supplement 2B—C*). Notably, expression of Notch1 and its target gene Hes1, as well as Notch ligand Dll4 and Jag2 were detected in these

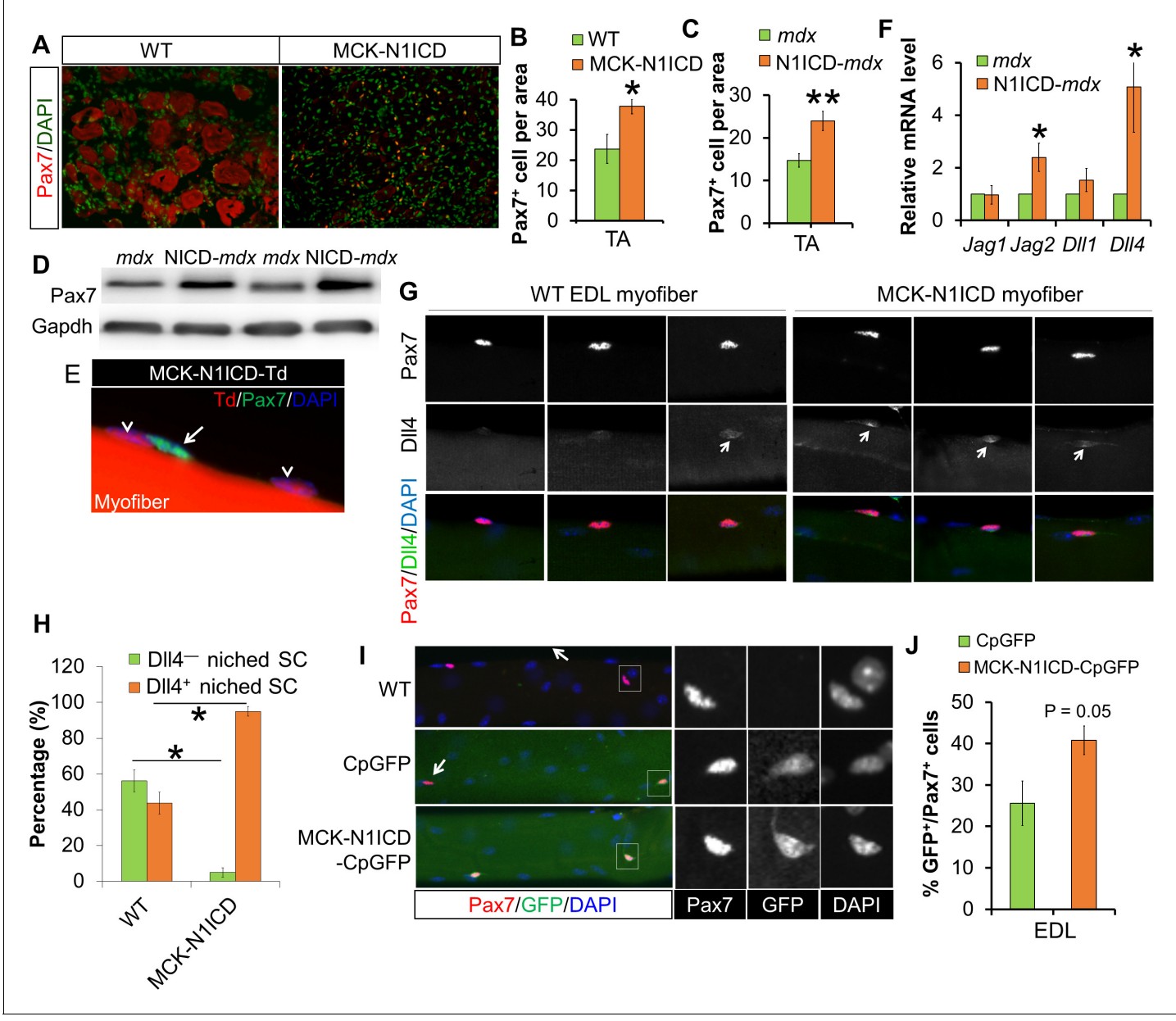

**Figure 6.** Notch activation in myofiber upregulates Notch ligands' expression, therefore stimulates Notch-activation and self-renewal of satellite cells niched on MCK-N1ICD myofibers. (A–C) Immunofluorescence images (A) and quantification of Pax7⁺ cell numbers in CTX-damaged (B, n = 3) and dystrophic TA muscles (C, n = 7). (D) Western blot results of Pax7 in muscle protein extracts. (E) Immunofluorescence image of one EDL myofiber to show that Pax7 cells are not from MCK-Cre lineage (Td ⁻) of MCK-N1ICD mice. Arrow points to Pax7⁺/Td⁻ cell, arrow-head points to Pax7⁻/Td⁺ myonucleus. (F) Relative mRNA levels of Notch ligand genes. (G,H) Immunofluorescence images (G) and quantification result (H, n = 3) of intact EDL myofibers. Arrow points to Dll4 immuno-signal patch on myofiber. (I,J) Immunofluorescence images (I) and quantification (J, n = 4) of GFP percentage in satellite cells (Pax7) on EDL myofibers. Arrow in I points GFP⁻/Pax7⁺ satellite cells; black and white images to shown individual channels of the outlined area on left images. *p<0.05, **p<0.01. Bar graphs indicate mean SEM.

The following figure supplements are available for figure 6:

**Figure supplement 1.** Real-time PCR results of Notch-related gene expression in primary myoblasts (A, n = 3) and myofibers (B, n = 5).

**Figure supplement 2.** Expression of Notch pathway genes in myofiber.

trypsin-stripped myofibers, though at lower levels compared to the non-trypsin-stripped myofibers (*Figure 6—figure supplement 2C—E*).

## Notch inhibition abolishes improvements in exercise performance and muscle regeneration in N1ICD-*mdx* mice

To directly test the hypothesis that Notch ligands on the MCK-N1ICD myofiber mediate SC function, we utilized dibenzazepine (DBZ), a γ-secretase inhibitor to block cleavage of N1ICD (thus ligand-induced Notch activation) without affecting the cleavage-independent N1ICD overexpression in myofibers. As myofibers are physically separated from the interstitial cells by basal lamina, Dll/Jag ligands on myofiber surface should only activate Notch receptors in juxtaposed SCs. Therefore, if the improved muscle function in MCK-N1ICD mice is mediated by elevated Notch ligand expression on myofiber, then the effect should be blocked by DBZ treatment.

We injected the mice with DBZ or vehicle control (DMSO) every two days continuously for 6 times, after which the mice were trained and tested on treadmill, and then examined for their regenerative capacity (*Figure 7A*). To test the long-term effect of modulating Notch signaling in SCs, we waited 20 days after last DBZ injection before phenotyping these mice. Whereas the N1ICD-*mdx* mice treated with DMSO ran faster and farther than their littermate *mdx* mice, after DBZ treatment both genotypes showed similar maximum speed and distance in the treadmill test (*Figure 7B and C*). In addition, DBZ-treated N1ICD-*mdx* muscle displayed similar EBD uptake levels to the *mdx* mice, indicating similar extent of muscle repair (*Figure 7D*). Indeed, H&E staining revealed better muscle regeneration in N1ICD-*mdx* than *mdx* mice in DMSO group, and DBZ administration rendered the regeneration similarly poor in both N1ICD-*mdx* and *mdx* mice (*Figure 7E*). Taken together, these results demonstrate that the amelioration of muscle atrophy and dysfunction in N1ICD-*mdx* mice is mediated by modulation of Notch signaling in SCs.

Collectively, our study demonstrates stage-specific functions of Notch during myogenesis. We show that N1ICD overexpression in committed and activated SCs, induced by *Myf5*[Cre] and *Myod1*-[Cre], respectively, completely blocked the embryonic muscle development (*Figure 7—figure supplement 1A*) (*Mourikis et al., 2012a*). *Myod1*[Cre]-N1ICD mice failed to survive to birth, and showed total absence of MyoD and MyoG expression (*Figure 7—figure supplement 1B and C*). However, MLC-N1ICD mice had relatively normal embryonic muscle development evidenced by similar body weight and muscle morphology at birth, but exhibited defective postnatal muscle growth and regeneration (*Figure 8*). Furthermore, adult MCK-N1ICD mice had normal muscle growth, and better muscle regeneration and motor function as they age (*Figure 8*). More importantly, MCK-N1ICD boosted muscle function and regenerative capacity in *mdx* mice (*Figure 8*). Together, our study revealed the versatile functions of Notch1 in controlling stepwise progression of myogenesis (*Figure 8*).

## Discussion

Notch signaling is indispensable for maintaining SC quiescence, and inactivation of Notch diminishes the SC pool (*Bjornson et al., 2012*; *Mourikis et al., 2012b*; *Wen et al., 2012*). Consistently, Notch signaling activity in the regenerating muscle correlates with quiescent stage of SC (*Bjornson et al., 2012*; *Mourikis et al., 2012b*). Although SCs can reversibly shift between quiescent and activated states, it's not clear whether post-differentiation myocyte or myofiber retains the potential to reverse back to the stem cell stage. Our study demonstrates that post-differentiation myocytes can dedifferentiate back into Pax7[+] quiescent SCs with activation of Notch1. Intriguingly, dedifferentiation was only observed in a portion of MLC-N1ICD cells, despite that N1ICD was overexpressed in both pre-fusion myocytes and post-fusion myotubes. By contrast, dedifferentiation was not observed in MCK-N1ICD mice, which specifically overexpress N1ICD in myotubes. Based on these comparisons, we concluded that Notch-induced dedifferentiation is restricted to mononucleated myocytes, which doesn't require cytokinesis as for multinucleated myofibers. Consistently, although knockout of Rb and Arf promoted cell cycle re-entry of myonuclei in myotubes, cytokinesis was not observed (*Pajcini et al., 2010*). One important future direction is to pinpoint the downstream targets of Notch1 that mediate its dedifferentiation function in myocytes.

Activation of Notch1 in myotubes improved motor function and muscle regeneration of both aged and *mdx* mice. This is accompanied by upregulation of Dll/Jag ligand expression in myofibers, higher Notch activity in juxtaposed SCs, as well as greater regenerative potentials of SCs in MCK-

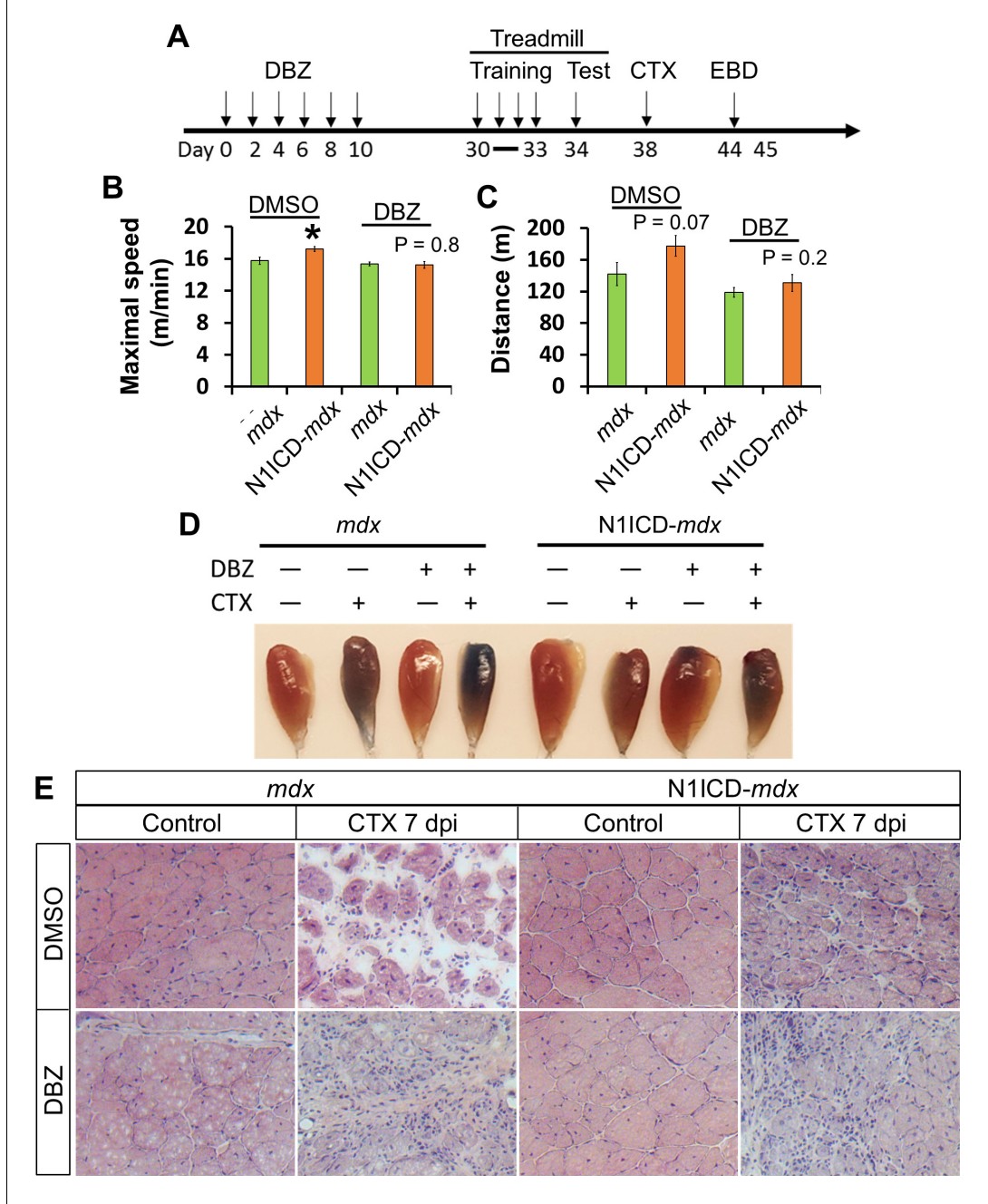

**Figure 7.** Notch inhibitor DBZ abolished improvements of exercise performance and muscle regeneration of N1ICD-*mdx* mice. (**A**) Experiment design to show the timing of different treatments. EBD, Evans blue dye. (**B,C**) Exhaustive Treadmill exercise test results (n = 6). (**D**) Image of EBD uptake by TA muscles. (**E**) H&E staining results of TA muscle cross sections. *p<0.05. Bar graphs indicate mean SEM.

The following figure supplement is available for figure 7:

**Figure supplement 1.** Absence of skeletal muscle development in *Myod1*[Cre]-NICD (MyoD-NICD) mice.

N1ICD mice. Consistently, Notch signaling deficiency in SCs is a prominent feature of aged and dystrophic muscles (*Conboy et al., 2003*; *Jiang et al., 2014*). As SCs are the only cells that are directly attached to the myofibers, Dll/Jag ligands on intact myofibers should only activate Notch signaling in the SCs, but not the other cells (for example blood vessel-associated cells and interstitial cells)

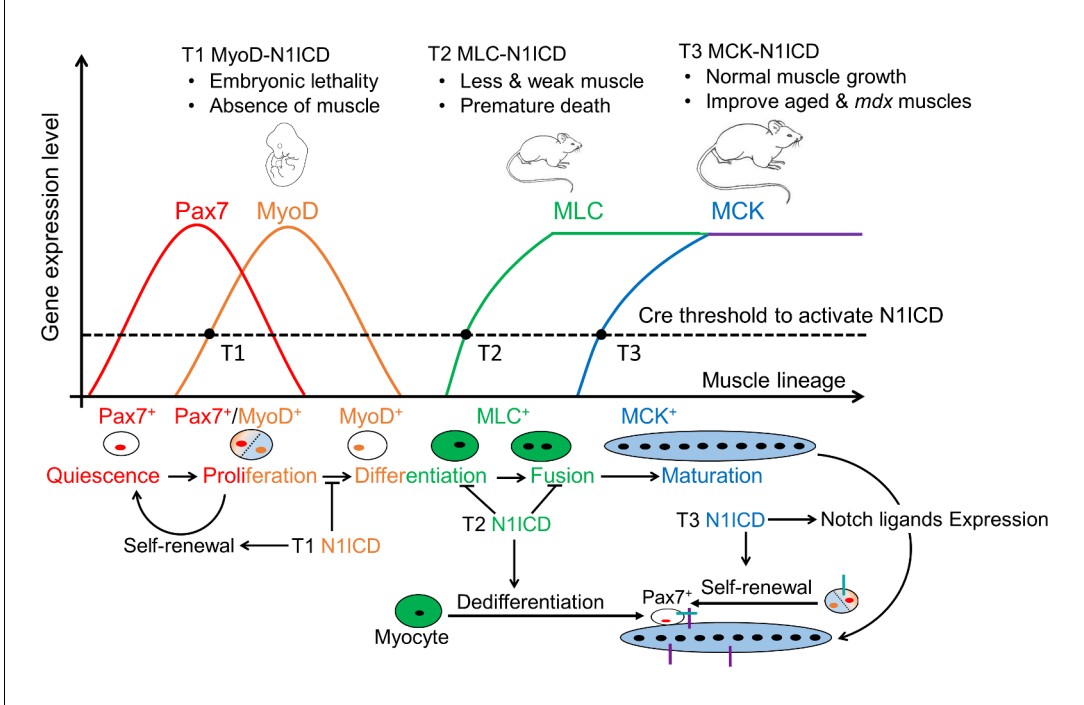

**Figure 8.** Summary of stage-dependent effects of Notch1 activation on muscle cell fate choice and myogenesis. Quiescent satellite cells (SCs) are marked as Pax7[+]. Expression of MyoD activates SCs, which enter into cell cycle. A subpopulation of the replicated SCs downregulate Pax7 to differentiate. After differentiation, MLC starts to express in myocytes and nascent myotubes, while MCK only starts to express in the multi-nucleated myofibers. Activation of Notch1 in MyoD[Cre] lineage (Timing 1, T1) blocks differentiation, promotes self-renewal of SCs, which causes absence of skeletal musculature and embryonic lethality; Activation of Notch1 in MLC[Cre] lineage (T2) induces dedifferentiation of myocytes, and generates Pax7 quiescent SCs. As a consequence, MLC-N1ICD mice show pronounced defects of muscle growth, motor-function and regeneration; Activation of Notch1 in MCK[Cre] lineage (T3) upregulates expression of Notch ligands on myofiber, which physiological promotes Notch activation in neighboring cells, inducing self-renewal of satellite cells. Thus, it improves muscle regeneration and exercise performance of old and *mdx* mice.

that are prevented from direct physical interaction with the myofiber by the basal lamina. Thus, the normalizing effects of DBZ on the improved muscle function in the MCK-N1ICD mice demonstrates that Notch-dependent enhancement of SCs' regenerative potential is the underlying molecular and cellular mechanism mediating the effect of N1ICD overexpression in myofibers. Although we detected in MCK-N1ICD muscle a gene expression profile similar to that of healthy human muscles, we didn't find any significant upregulation of Notch pathway genes in healthy human subjects. Considering the multiple cell types in muscle tissues, expression and function of Notch in individual cell types warrant further investigation.

In summary, our results illustrate contrasting stage-dependent effects of Notch1 activation in post-differentiation muscle cells. Most importantly, our study demonstrates that activation of Notch1 improves myotube's function as a stem cell niche. This finding suggests that one strategy to ameliorate sarcopenia and muscle dystrophy diseases is through modulating Notch signaling in the myofiber.

# Materials and methods

## Animals

All procedures involving mice were approved by Purdue University's Animal Care and Use Committee under protocol # 1112000440. *Myl1*[Cre] mice were generously provided by Steven Burden (Skirball Institute of Biomolecular Medicine, NYU). The other mice were purchased from Jackson lab: Ckmm-Cre, also known as MCK-Cre (stock# 006475), *Rosa26(Gt)Sor*[N1ICD] (stock# 008159), *Rosa26(Gt)Sor*[td-Tomato] (stock# 007914), *Rosa26(Gt)Sor*[nTnG] (stock# 023035), *mdx* (stock# 001801), CpGFP

(stock#005854). Mice were housed in the animal facility with free access to water and standard rodent chow food.

## Muscle injury and regeneration

Muscle regeneration was induced by injection of cardiotoxin (CTX; Sigma-Aldrich, St. Louis, MO). Briefly, mice were anesthetized using a ketamine-xylazine cocktail, and then 50 µl of 10 µM CTX was injected into the TA muscle. Muscles were then harvested at indicated time post-injection to assess the completion of regeneration and repair.

## DBZ treatment

DBZ was purchased from TOCRIS Bioscience (catalog number 4489), and dissolved in DMSO at a 100 mM concentration. Before use, the stock was suspended at a 1:200 dilution in DMSO. 50 µl working solution was injected into each TA muscle of mice. Mice were anesthetized using a keta-mine-xylazine cocktail before DBZ injection.

## Myosin heavy chain isoforms separation by SDS-PAGE

MHC isoforms were separated by modified SDS-PAGE (8% acrylamide gels containing 3.5% glyc-erol), which was run for 22 hr at 4°C. MHC isoforms were detected by silver stain after gel fixation.

## Treadmill and gripping strength measurement

A detailed protocol for treadmill and grip strength tests of mice are published elsewhere (*Castro and Kuang, 2017*). In brief, the mouse was trained on treadmill (Eco3/6 treadmill; Columbus Instruments, Columbus, OH) with a fixed 10% slope, at constant 10 m/minute speed for 5 min daily for consecutively 3 days before test. On the exercise testing day, the animals ran on the treadmill at 10 m/min for five minutes and the speed was increased by 2 m/min every two minutes until they were exhausted or a maximal speed of 46 m/min was achieved. The exhaustion was defined as the inability of the animal to run on the treadmill for 10 s despite mechanical prodding. Running time and maximum speed achieved was measured whereas running distance was calculated. A digital grip-strength meter (Columbus Instruments) was used to measure four-limb grip strength in mice, which were acclimatized for 10 min before the test. The mouse was allowed to grab the metal pull bar with the paws. Then the mouse tail was gently pulled backward until mice could not hold on the bar. The force at the time of release was recorded as the peak tension. Each mouse was tested three times. The average strength was defined as grip strength.

## Denervation and neuromuscular junction (NMJ) staining

For sciatic nerve denervation, mice were anesthetized with ketamine. The right hind limb was pre-pared for surgery, a 1-cm incision was made in the skin along with the axis of the femur, and the sci-atic nerve was isolated. To prevent reinnervation, 3–5 mm section of the sciatic nerve was cut and removed. Mice were sacrificed by cervical dislocation 2 weeks after denervation. For NMJ staining, muscles were dissected and fixed by 4% PFA and incubated 30% sucrose/PBS for overnight. Muscles were frozen by cold isopentan with OCT-compound and sectioned longitudinally at 50 µm thickness. Sections were incubated with blocking buffer containing 0.1% Triton-X 100 and biotinylated α-bun-garotoxin (α-BTX) to detect NMJs for overnight. NMJs were visualized by FITC-conjugated streptavidin.

## Muscle myoblast isolation and culture

Primary myoblasts were isolated with collagenase type I and dispase B digestion. Briefly, the hind limb skeletal muscles of mice were collected, minced and digested for around 40 min. The diges-tions were stopped with F-10 Ham's medium containing 10% FBS and centrifuged at 450 ×g for 5 min. Then the cells were seeded on collagen-coated dishes and cultured in growth medium contain-ing F-10 Ham's medium, with 20% fetal bovine serum (FBS), 4 ng/mL basic fibroblast growth factor, and 1% penicillin–streptomycin at 37°C with 5% $CO_2$. The medium was changed every 2 days.

## Single myofiber isolation and immunostaining

Single myofiber was isolated from the extensor digitorum longus (EDL) muscle by digestion with 0.2% collagenase A (Sigma) in DMEM for around 45 min. After digestion, EDL muscle was gentally titrated by different pore-sized glass pipettes. Freshly isolated single fiber was quickly fixed in 4% PFA for 10 min, washed in 100 mM glycine three times each 10 min, followed by PBS wash for three times and blocking for 1 hr at room temperature. The primary antibody was incubated at 4°C for overnight. Fluorescence secondary antibody was diluted and incubated at room temperature for 1 hr. Images were taken with a Leica DM6000 microscope with a 20× objective.

To strip SCs from myofiber and get SC-free myofibers, freshly isolated EDL myofibers were first washed with PBS for three times. Myofibers were trypsinized with 0.25% trypsin for 5 mins to release SCs and other cells, and then washed with PBS for three times. Myofibers without trypsin treatment were used as a control for immunostaining or gene expression analysis. For gene expression analysis, mRNA from all myofibers liberated an EDL muscles was extracted (~100 myofibers) and reverse transcribed, and subjected to PCR analysis. PCR cycle number used for 18 s was 30, and for Notch1, Pax7, Dll4 and Jag2 was 38.

## Hematoxylin-eosin (H&E) and immunofluorescence staining

A detailed protocol for muscle histological characterization using H&E staining and myofiber typing using immunofluorescence are published in Bio-Protocol (*Wang et al., 2017*). Briefly, fresh TA muscles were embedded in OCT compound, and frozen in isopentane chilled on dry ice. Then the tissues were cut at 10 μm thickness by Leica CM1850 cryostat. Muscles sections were fixed with fresh 4% PFA for 10 min, washed in 100 mM glycine three times each 10 min, followed by PBS wash for three times and blocking for 1 hr at room temperature. The primary antibody was incubated at 4°C for overnight. Fluorescence secondary antibody was diluted and incubated at room temperature for 1 hr. Images were taken with a Leica DM6000 microscope with a 20× objective. H&E staining images were captured with a Nikon D90 digital camera mounted on a microscope.

## Protein extraction and Western blot analysis

Protein was isolated from cells or tissue using RIPA buffer contains 50 mM Tris-HCl (pH 8.0), 150 mM NaCl, 1% NP-40, 0.5% sodium deoxycholate and 0.1% SDS. Protein concentrations were determined using Pierce BCA Protein Assay Reagent (Pierce Biotechnology), followed by measurement with NanoDrop. Proteins were separated by sodium dodecyl sulfate polyacrylamide gel electrophoresis (SDS-PAGE), transferred to a polyvinylidene fluoride (PVDF) membrane (Millipore Corporation), blocked in 5% fat-free milk for one hour at room temperature, and then incubated with primary antibodies diluted in 5% milk overnight at 4°C. The Hes1 (sc), Hey1 (sc), Gapdh (sc-32233) antibodies were from Santa Cruz Biotechnology. The HRP conjugated secondary antibody (anti-rabbit IgG or anti-mouse IgG, Santa Cruz Biotechnology) was diluted at 1:5000. Immunodetection was performed using enhanced chemiluminescence (ECL) Western blotting substrate (Pierce Biotechnology) and detected with an imaging system (FluorChem R FR0116). Alternatively, the membrane was incubated with an infrared secondary antibody (Alexa Fluor 790 goat anti-mouse IgG, A11357; Alexa Fluor 680 goat anti-rabbit IgG, A21109; Life Technologies, USA) diluted 1:10,000, and the signals were detected by using the Odyssey infrared image scanning system.

## Total RNA extraction, cDNA synthesis, real-time PCR

Total RNA was extracted from cells or tissues using Trizol Reagent according to the manufacturer's instructions. The purity and concentration of total RNA were determined by a spectrophotometer (Nanodrop 2000c, Thermo Fisher) at 260 nm and 280 nm. Ratios of absorption (260/280 nm) of all samples were between 1.8 and 2.0. Then 5 μg of total RNA were reverse transcribed using random primers with M-MLV reverse transcriptase (Invitrogen). Real-time PCR was carried out in a Roche Light Cycler 480 PCR System with SYBR Green Master Mix (Roche) and gene-specific primers as previously described (*Liu et al., 2012b*). The $2^{-\Delta\Delta Ct}$ method was used to analyze the relative changes in each gene's expression normalized against 18S rRNA expression.

## Microarray, IPA and GSEA analysis

RNA was extracted from TA muscles of adult *mdx* and MCK-N1ICD-*mdx* mice. Gene expression was analyzed by microarray with Agilent SurePrint G3 Mouse GE 8 X 60 K chip. The list of significantly changed genes with a fold change ≥1.5–fold was used for GSEA analysis, as well as analyzed through the QIAGEN's Ingenuity Pathway Analysis (IPA, QIAGEN). GSEA analysis was performed as instructed by the manual with default settings of software.

## Statistics

Trial experiments or experiments done previously were used to determine sample size with adequate statistical power. Measurement values that are beyond the fence as determined by interquartile range were considered as outlier and excluded from following statistical analysis. All analyses were conducted with student *t* test with two-tail distribution. Comparison with a *p* value <0.05 was considered significant.

## Acknowledgements

We thank Dr. Shuichi Sato and Dr. Ashok Kumar for assistance with treadmill experiments; Jun Wu for maintaining mouse colonies; and members of the Kuang laboratory for comments. This work was partially supported by a grant from the US National Institutes of Health (R01AR060652).

## Additional information

### Funding

| Funder | Grant reference number | Author |
|---|---|---|
| National Institute of Arthritis and Musculoskeletal and Skin Diseases | AR060652 | Shihuan Kuang |

The funders had no role in study design, data collection and interpretation, or the decision to submit the work for publication.

### Author contributions

PB, Approved the manuscript, Conception and design, Acquisition of data, Analysis and interpretation of data, Drafting or revising the article; FY, YS, SW, WL, TS, YW, DZ, JF, Approved the manuscript, Acquisition of data, Analysis and interpretation of data; SK, Approved the manuscript, Conception and design, Analysis and interpretation of data, Drafting or revising the article

### Author ORCIDs

Yefei Wen, http://orcid.org/0000-0002-2121-7538
Shihuan Kuang, http://orcid.org/0000-0001-9180-3180

### Ethics

Animal experimentation: All procedures involving mice were approved by Purdue University's Animal Care and Use Committee under protocol # 1112000440.

## Additional files

### Major datasets

The following previously published dataset was used:

| Author(s) | Year | Dataset title | Dataset URL | Database, license, and accessibility information |
|---|---|---|---|---|
| Pescatori M, Broccolini A, Minetti C, | 2007 | Early-early stage Duchenne muscular dystrophy: quadriceps | http://www.ncbi.nlm.nih.gov/sites/GDSbrowser? | Publicly available at the NCBI Gene |

| Bertini E et al | acc=GDS3027 | Expression Omnibus (Accession no: GDS3027) |

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
