## [Decision Letter]

Thank you for submitting your article "Stage-specific functions of Notch activity during skeletal myogenesis" for consideration by *eLife*. Your article has been reviewed by two peer reviewers, and the evaluation has been overseen by a Reviewing Editor and Fiona Watt as the Senior Editor. The reviewers have opted to remain anonymous.

The reviewers have discussed the reviews with one another and the Reviewing Editor has drafted this decision to help you prepare a revised submission.

Summary:

In this report the authors have taken a genetic approach to analyse the impact of ectopic Notch signalling activation during muscle differentiation. In the first part of the study, MLC-Cre is used to drive the expression of constitutively activated Notch1 (N1ICD) in cells engaged to differentiation. MLC-N1ICD muscles are shown to have an impressive increase of non-cycling, Pax7 cells, as a consequence of a conversion of differentiating myoblasts to Progenitor/satellite cell-like cells. In the second part, MCK-Cre is used to induce N1ICD in the formed myofibers. Surprisingly, this results in improved muscle regeneration of normal and *mdx* mutant muscle, suggesting a non-cell autonomous mechanism, involving the muscle fibers and the satellite cells. Also, MCK-N1ICD-*mdx* mice show improved performance on a treadmill compared to *mdx* mice. Moreover, six injections of a Notch inhibitor DBZ to MCK-N1ICD-*mdx* mice are sufficient to revert the phenotypes to control levels in animals tested 20 days after the last DBZ injection. This striking result led the authors to propose that the interaction between the satellite cells and the fiber is mediated by Notch signalling, as the MCK-Nicd fibers express high levels of the Notch ligands Dll-4 and Jag2.

It is well known that the notch signaling pathway decreases during aging in muscle but the role of notch in the myofiber and its impact upon muscle regeneration has never been addressed. The authors have brought new insights regarding the molecular mechanisms leading to satellite cell self-renewal through the notch signaling pathway. They also highlight the communication inside the muscle satellite cell stem cell niche, i.e. between myofibers and satellite cells. Experiments have been well conducted and authors' conclusions are consistent with their results.

Essential revisions:

No information is provided on the endogenous activity of Notch is the systems analysed. In fact, previous work has shown that Notch activity declines as cells differentiate. The authors could measure Notch activity and the expression of its components in more detail to provide a clear interpretation of the data; Is there any endogenous Notch activity in the fiber? Does the fiber normally express Jag2 and Dll4?

All transcriptional analyses are performed with whole muscles extracts, including the microarrays. Candidate targets should at least be validated using purified cells: satellite cells, myoblasts and muscle fibers.

The authors uncover an impressive Pax7 phenotype in the MLC-Nicd mice. Moreover, these Pax7 cells are negative for Ki67 and Myogenin, suggesting that they are quiescent. It would be informative to stain these cells also for Myod and Myf5 (although Myf5 antibody is not easy to work) in order to understand if the Nicd-expressing cells are satellite cell-like, are blocked at another intermediate state.

The observation that these cells are Ki67 (or other proliferation marker) is key to conclude that these cells are quiescent. The authors should stain with Ki67 regenerating muscles of Control and MLC-Nicd mice. Control muscles would serve as a control for the Ki67 staining, which is needed for the negative result of Figure 3. In parallel, it would be interesting to assess the cycling state of the Nicd-expressing cells during regeneration. This would also be particularly relevant, as work from the Kuang lab (Wen Y. et al., MCB 2012) has previously shown that NICD overexpression inhibits S-phase entry and Ki67 expression.

---

## [Author Response]

*No information is provided on the endogenous activity of Notch is the systems analysed. In fact, previous work has shown that Notch activity declines as cells differentiate. The authors could measure Notch activity and the expression of its components in more detail to provide a clear interpretation of the data; Is there any endogenous Notch activity in the fiber? Does the fiber normally express Jag2 and Dll4?*

To examine the endogenous expression of Notch pathway genes in myofibers, we attempted to isolate EDL myofibers free of any other cell contamination. EDL muscles were digested in collagenase and dispase to dissociate myofibers, single myofibers were further digested with trypsin (plus multiple rounds of PBS wash) to strip any cells that might have attached to the myofiber surface, such as satellite cells, blood capillary cells and interstitial cells (Figure 6—figure supplement 2). As a proof of concept, we show by direct visualization (using the Pax7^nGFP^ reporter) and gene expression in Figure 6—figure supplement 2 that the trypsin-stripped myofibers are free of satellite cells. As satellite cells are beneath the basal lamina and any other myofiber-associated cells are located outside of the basal lamina, the lack of satellite cells indicates that all myofiber-associated cells should have been removed from the trypsin-stripped myofibers.

We then compared the gene expression of EDL myofibers before and after trypsin stripping. We detected robust expression of Notch1, Dll4 and Jag2 in the trypsin-stripped myofibers, though at slightly lower levels compared to the non-trypsin-stripped myofibers (Figure 6—figure supplement 2). Immunostaining and western blot also confirmed expression of Dll4, Notch1, Hes1 in the stripped myofibers (Figure 6—figure supplement 2). The expression of Hes1, a common target of Notch signaling, further demonstrates that not only is Notch1 receptor expressed, but the Notch1 receptor is also activated in the myofiber.

*All transcriptional analyses are performed with whole muscles extracts, including the microarrays. Candidate targets should at least be validated using purified cells: satellite cells, myoblasts and muscle fibers.*

To address the reviewer’s concern, we analyzed expression of Notch1, Dll4, Jag2 and Notch target genes in myofibers to verify our previous results, as they composed the most important mechanistic part of this paper. As shown in Figure 6—figure supplement 1, we consistently detected the higher expression levels of these genes in freshly isolated, stripped MCK-N1ICD myofibers, when compared with WT myofibers. We have also compared myoblasts isolated from WT and MCK-NICD mice, and detected upregulation of Hes5 and Hey2 (Figure 6—figure supplement 1). Plus, the Cp-GFP Notch reporter showing higher abundance of Notch-on satellite cells in the MCK-NICD muscles compared to WT muscles (Figure 6), we hope these results will satisfy the reviewers.

*The authors uncover an impressive Pax7 phenotype in the MLC-Nicd mice. Moreover, these* Pax7*cells are negative for Ki67 and Myogenin, suggesting that they are quiescent. It would be informative to stain these cells also for Myod and Myf5 (although Myf5 antibody is not easy to work) in order to understand if the Nicd-expressing cells are satellite cell-like, are blocked at another intermediate state.*

As suggested by the reviewer, co-staining of Pax7 and MyoD has been widely used to report the status of satellite cells. Specifically, quiescent, activated and differentiating satellite cells can be identified as Pax7 /MyoD^—^, Pax7 /MyoD and Pax7^—^/MyoD. Consistent with our previous gene expression analysis, all the Pax7 cells in MLC-N1ICD muscles were negative for MyoD (Figure 3—figure supplement 2). This result validates the quiescent state of these cells. As a positive control of staining, cultured WT myoblast were Pax7 ^/^MyoD (Figure 3—figure supplement 2).

*The observation that these cells are Ki67 (or other proliferation marker) is key to conclude that these cells are quiescent. The authors should stain with Ki67 regenerating muscles of Control and MLC-Nicd mice. Control muscles would serve as a control for the Ki67 staining, which is needed for the negative result of Figure 3. In parallel, it would be interesting to assess the cycling state of the Nicd-expressing cells during regeneration. This would also be particularly relevant, as work from the Kuang lab (Wen Y. et al., MCB 2012) has previously shown that NICD overexpression inhibits S-phase entry and Ki67 expression.*

As suggested by the reviewer, we stained Ki67 for both freshly isolated myofibers and regenerating muscles (7 dpi) of MLC-N1ICD mice. Consistently, we didn’t detect any Ki67 cells on fresh isolated MLC/N1ICD myofibers (Figure 3—figure supplement 2). Also, while roughly 50% of the Pax7 cells were stained as Ki67 in regenerating WT muscles, only 15% Pax7 were Ki67 in regenerating MLC-N1ICD muscles (Figure 3—figure supplement 2).